# Mid-winter (DJF) temperature reconstruction in

# Jerusalem since 1750 with some regional implications

Assaf Hochman<sup>1, 2, 5</sup>, Hadas Saaroni<sup>2</sup>, Miryam Bar-Matthews<sup>3</sup>, Baruch Ziv<sup>4</sup>, and

Pinhas Alpert<sup>1</sup>

<sup>1</sup>Department of Earth Sciences, Tel-Aviv University, Tel-Aviv, Israel.

<sup>2</sup>Department of Geography and the Human Environment, Tel-Aviv University, Tel-Aviv, Israel.

<sup>3</sup>Geological Survey of Israel, Jerusalem, Israel.

<sup>4</sup>Department of Natural Sciences, the Open University of Israel, Ra'nana, Israel.

<sup>5</sup> Porter School of Environmental Studies, Tel-Aviv University, Tel-Aviv, Israel.

Submitted to: Climate of the Past on September 2016

## **Correspondence author:**

Assaf Hochman

E -Mail assafhochman@yahoo.com

### 1 Abstract

| 2  | This work presents a statistical reconstruction of average mid-winter (DJF)                 |  |  |
|----|---------------------------------------------------------------------------------------------|--|--|
| 3  | temperature in Jerusalem since 1750. It is a first comprehensive attempt to reconstruct     |  |  |
| 4  | the temperature in Jerusalem, as a good representation of the Eastern Mediterranean         |  |  |
| 5  | (EM) climate. This representativeness is verified here. The data has been reconstructed     |  |  |
| 6  | by using a statistical model based on Principal Component Regression (PCR), using           |  |  |
| 7  | both instrumental data and high temporal resolution records of proxy data, including        |  |  |
| 8  | tree ring chronologies from Jordan, and records of DJF precipitation and Sea Level          |  |  |
| 9  | Pressure from central and Western Europe. A split validation procedure has resulted in      |  |  |
| 10 | a 0.73 correlation between observed and reconstructed temperature. The warming trend        |  |  |
| 11 | of last decades is well noted in the reconstruction and is in line with other studies.      |  |  |
| 12 | Winters which were cold/warm were historically documented as wet/dry, respectively,         |  |  |
| 13 | consistent with earlier studies pointing a strong relationship between Jerusalem            |  |  |
| 14 | temperatures and precipitation. It is shown here for the first time that the 'First Aliyah' |  |  |
| 15 | (immigration) to Israel during 1882-1904 initiated during favouring climate conditions      |  |  |
| 16 | (cool and wet) to establish an agricultural community in the region. These conditions       |  |  |
| 17 | were found to be exceptional compared to other periods since 1750.                          |  |  |
| 18 |                                                                                             |  |  |
| 19 | Keywords: Climate reconstruction; Principal Component Analysis; climate change;             |  |  |
| 20 | First Aliyah; immigration                                                                   |  |  |
| 21 |                                                                                             |  |  |
| 22 |                                                                                             |  |  |
| 23 |                                                                                             |  |  |
| 24 |                                                                                             |  |  |
| 25 |                                                                                             |  |  |
| 26 |                                                                                             |  |  |
| 27 |                                                                                             |  |  |
| 28 |                                                                                             |  |  |
| 29 |                                                                                             |  |  |
| 30 |                                                                                             |  |  |

## 32 **1 Introduction**

Temperature and precipitation are the most important climatic elements affecting 33 34 humanity, economy and ecosystems. Warming and drying trends, as well as extreme temperature events, including heat waves, dry spells and droughts have an essential 35 36 influence on human life, especially in the Eastern Mediterranean (EM) region (Saaroni et al., 2003; Luterbacher et al., 2004; Touchan et al., 2005; Ziv et al., 2005; Luterbacher 37 38 et al., 2012; Lelieveld et al., 2012; Tanarhte et al., 2012; Saaroni et al., 2015). Several studies suggested that climate change had a decisive impact on societies in the region, 39 at times even bringing about total collapse (Alpert and Neumann, 1989; Weiss and 40 Bradley, 2001; Enzel et al., 2003; Ellenblum, 2012; Torfstein et al., 2013, Kelley et al., 41 2015). For example, recent studies suggest that Syria and the greater Fertile Crescent, 42 43 where agriculture had begun some 12,000 years ago, experienced the worst 3-year drought in the instrumental record (Trigo et al., 2010). The drought caused massive 44 agricultural failures. The most significant consequence was the migration of as many 45 as 1.5 million people from rural farming areas to urban centers. It was further suggested 46 47 that the drought (2007-2010), contributed to the present conflict in Syria (Kelley et al., 2015). The first study to point at global warming as a central contributor to the drying 48 49 of the Fertile Crescent was Kitoh et al., (2008), see also Alpert et al., (2013).

The EM region has received relatively little attention concerning past multi proxy 51 high resolution climate reconstructions (Finne et al., 2011; Luterbacher et al., 2012; Lelieveld et al., 2012) despite the unique position of the region, as a transition zone 52 between temperate climate in the North and arid climate in the South (Bar-Matthews et 53 al., 1997; Lionello et al., 2014). The EM climate is influenced by subtropical and mid-54 55 latitude circulations as well as by tropical intrusions. Fluctuations in the occurrence and intensity of these circulations are known to be controlled by different large-scale 56 oscillations; This leads to complex features in the region's climatic fluctuations (Alpert 57 et al., 2006; Saaroni et al., 2010). 58

## 60 1.1 Recent studies of past climate in the EM

The use of proxy data from natural climate archives in climate reconstruction helps to expand the time scale of the study, and enables investigation of multi-decadal variations (Luterbacher et al., 2012). Various publications deal with European temperature reconstruction at different scales of time and space (Luterbacher et al., 1999; Luterbacher et al., 2004; Casty et al., 2005a; Xoplaki et al., 2005). In Israel and

nearby areas a few attempts have been made at climate reconstruction with high 66 resolution data, covering the past centuries. Liphschitz and Waisel, (1967) used tree 67 ring data to develop paleo-climate records of the region. Tamari, (1976) has 68 reconstructed rainfall regimes in Northern Sinai and Southern Israel through Dendro-69 chronological analysis since 1850. Frumkin et al., (1991), Bar-Matthews et al., (1997) 70 and Vaks et al., (2003) have shown isotopic fractionation analysis of speleothems in 71 72 karstic caves in Israel to infer precipitation at the caves sites. Kuperman, (2005) had reconstructed flood periods during the Holocene by dating speleothems in the Northern 73 74 Negev, Israel. Touchan and Hughes, (1999) have reconstructed rainfall regimes in Southern Jordan since 1600 using Dendro-chronological analysis. Enzel et al., (2003) 75 76 exposed the surface levels of the Dead Sea in a resolution of decades to centuries, using 77 it as a proxy for winter rainfall in the region. In a recent study, droughts were reconstructed using Tree-ring chronologies from the Mediterranean region (Cook et al., 78 79 2016). They suggested that the recent drought in the Eastern Mediterranean is the worst 80 in 900 years.

No comprehensive attempt has been made to reconstruct the regional climate of Jerusalem, by combining and integrating different proxies, with high temporally 82 83 resolved records. The various proxies differ widely in their temporal resolutions. Some proxy records have annual or longer time scales. Several works (Pauling et al., 2003; 84 85 Luterbacher et al., 2012) have reviewed the strengths and potential weaknesses of proxy sources used for seasonal climate reconstructions in the Mediterranean region. They 86 concluded that an optimal combination of high quality natural and documentary proxies 87 can yield better results than using only one of the two types. 88

#### 90 **1.2 The climate of Jerusalem**

Jerusalem is a town located on the Judean Mountains, at an elevation of about 600-800 m. It is situated on the border between temperate climate to its north and west and 92 93 the semi-arid and arid climate to its east, known as the Judean Desert, which is a 'rain 94 shadow' desert. Jerusalem belongs to the Mediterranean climatic region (according to the Köppen classification) with an average annual precipitation of 537 mm. 95 96 Precipitation occurs mostly during Oct-May. January is the coldest month of the year 97 with an average temperature of 9.8°C; July and August are the hottest months with and average temperature of 25°C. Temperatures vary widely from day to night (IMS 2016). 98

This study aims to reconstruct the mid-winter (DJF) temperature fluctuation in
Jerusalem since 1750. The methodology uses a statistical model based on Principal
Component Regression (PCR), combining instrumental data and different documentary
and natural proxies.

## 104 **2 Data**

We draw on homogenised mid-winter (DJF) mean temperature of Jerusalem for the period of 1865-2015, as the target of our model; retrieved from the Global Historical Climatology Network (GHCN-adjusted, Lawrimore et al., 2011) and verified as homogenised at the Israel Meteorological Service. Our analysis also compares this record with the precipitation record of Jerusalem for the same period.

Our preliminary data base consisted of 49 long time-series of temperature, precipitation, and Sea Level Pressure (SLP) measurements taken in central and Western Europe, retrieved from the GHCN adjusted station data (Lawrimore et al., 2011). Most of these series start in the mid-18<sup>th</sup> century with virtually no missing values; some series go further back to the end of the 17<sup>th</sup> century. The relevance of the above data for the reconstruction of temperature in Jerusalem is established in section 2.3 (Figure 1).

In addition, ten tree-ring site chronologies originating in the Dana Reserve in Jordan and in the Troodos Mountains in Cyprus (Table 1) (www.ncdc.noaa.gov) underwent de-trending and standardization by ARSTAN software (Cook, 1985) in order to eliminate an age trend without losing high and low frequency climatic signals. Furthermore, we have used historically documented data from Jerusalem, since 1750 (Klein, 1985). Brazdil et al., (2005) have suggested that this kind of documentary source can be used to pin point extremes.

The investigation of the synoptic and large-scale patterns that affect Jerusalem was made possible through the use of NCEP/NCAR reanalysis archive (Kalnay et al., 1996; Kistler et al., 2001), along with Kaplan's Global Sea Surface Temperature (SST) reconstructions (<u>http://climexp.knmi.nl</u>). All in all we started with 59 original timeseries, which make the full data base. Figure 1 displays a map with the origin of each data source described in this section.

# 133 **3 Statistical reconstruction model**

Our statistical modelling approach consists of three steps:

Reduction of dimensionality – first choosing significant correlated
 predictors with Jerusalem mean DJF temperature and then Principal Component
 Analysis (PCA) on the chosen predictors.

2) Temperature reconstruction – Multiple Linear Regression (MLR).

3) Validation – Split Validation (SV).

PCA is a common technique used in statistical modelling, where the number of 140 predictors used in the computation is reduced for the sake of a more robust model. PCA 141 produces artificial predictors, which are uncorrelated to one another, in order to avoid 142 ambiguity in the climatic signal (Mierswa et al., 2006). In this study, significant 143 144 correlated predictors (95% significant level) to the mean DJF temperature in Jerusalem were found. Before applying PCA we normalised the various proxies in order to avoid 145 dominance of numerically larger attributes over smaller ones (Hsu et al., 2003). Next, 146 PCA was performed, keeping 70% of the variance in the original data while retaining 147 148 the least number of principal components. The chosen PCs were learned by a MLR model, using the period 1865-1981, for which that proxies can be related to Jerusalem 149 150 temperature with virtually no missing values. The model produced was applied on the reconstructed period (1750-1864). 151

Validation of the reconstruction was done by Split Validation, sometimes called rotation estimation, which is the statistical practice of partitioning a sample of data into subsets so that analysis is initially performed on a single subset while the rest of the set is retained for confirmation of the initial analysis (Michaelson, 1987). The initial subset is the training set and the other is validation or testing sets. In this study, the data of 1865-1981 were randomly partitioned into a training set and a validation set.

138

#### 159 **4 Results**

#### 160 4.1 Jerusalem as a representative of Eastern Mediterranean climate

Studies have shown that the Jerusalem precipitation record is a reliable proxy for 162 the hydro-climatic fluctuations in the EM (Enzel et al., 2003, Kushnir and Stein, 2010). 163 Following this, we applied two tests to evaluate how well the recorded temperature in 164 Jerusalem represents the temperature of the EM. First, we evaluated the coherence 165 among the nine longest records of instrumental surface temperature from stations 166 located in the EM; Jerusalem (Israel), Beirut (Lebanon), Amman (Jordan), Nicosia

(Cyprus), Port-Said (Egypt), Abbasiya (Egypt), Antalya (Turkey), Urfa (Turkey) and Adana (Turkey). Using correlation coefficients between every two pairs of these 168 records, each station received a score of "representativeness". The score of each station 169 170 was calculated as the ratio between the average of all correlation coefficients it has and 171 the average of the significance values for these correlations. This way, a higher average correlation or a lower average significance both contribute to a higher overall score. 172 173 Jerusalem turned out to be the most representative of them, followed by Nicosia and 174 Amman (Table 2).

The second criterion to evaluate the representativeness of Jerusalem was displaying a comparison of the spatial correlation between the average temperature in Jerusalem 176 and GHCN surface temperature fields in other places in the region and in Europe for 177 DJF, during the period of 1900-1990 (Figure 2). The resulting high significant 178 correlations with other locations in the EM, lead us again to conclude that Jerusalem is 179 a suitable representative for the climate of the region. Accordingly, the reconstruction 180 181 of temperature variations in Jerusalem is an important first step towards a better 182 understanding of climatic variability in the EM.

To justify our use of both temperature, precipitation and SLP data in Central Europe 183 184 as good predictors for the temperature in Jerusalem, we analysed the relationship between these variables during the 20th century, using correlation maps, to identify tele-185 186 connection patterns. Figure 2 displays correlation maps between mid-winter (DJF) temperature in Jerusalem and temperature, precipitation and SLP in Europe for the same 187 period. The results indicate a strong relationship between the values in the two regions: 188 a positive correlation pattern between precipitation in south Europe and mean 189 190 temperature in Jerusalem, and a negative one between temperature/SLP in Europe and 191 mean temperature in Jerusalem.

These findings suggest that connection exists between mid-winter (DJF) temperature 192 in Jerusalem and large-scale circulation patterns. Figure 3 presents teleconnection 193 194 patterns between Jerusalem mid-winter temperature and SST's. A negative relationship 195 (R<-0.3) between SST in the North Sea and a positive (R>0.4) with the tropical Atlantic and Jerusalem temperature is observed. Figure 4 presents teleconnection patterns 196 197 between Jerusalem mid-winter temperature and 300mb geopotential heights. A strong negative relationship is observed over central and western Europe. These findings along 198 with the negative SLP relationship with Jerusalem mid-winter temperature (Figure 2) 199 200 reinforce the statistical relationship between mid-winter precipitation in Europe and

- temperature in Jerusalem, with a dynamical one. European high pressure causesnortherly winds in Israel which bring cold air from western Russia to the region, in a
- process governed by the phase configuration of Rossby waves (Ziv et al., 2006).

## 205 4.2 The Statistical reconstruction model and validation results

Table 3 shows the final predictors found to be significantly (95% level) correlated to Jerusalem mid- winter (DJF) temperature. 70% of variance in the data was kept by the first **five Principle Components (PC)**, while the first PC explained **41%** of the variance. The final regression equation for Jerusalem mid-winter (DJF) temperature is displayed: 0.25\*PC1 + 0.16\*PC2 + 0.06\*PC3 - 0.11\*PC4 + 0.18\*PC5 + 8.45

The split validation has resulted in relatively good performance; with 0.73 correlation coefficient (Table 4 displays performance indicators for the reconstruction model). Figure 5 displays the ability of the statistical model to reconstruct the average winter temperature of Jerusalem. It is shown that the model can simulate relatively well the mid-winter (DJF) temperature fluctuations in Jerusalem.

#### 217 **4.3 Reconstruction results**

Final results for the reconstruction of normalised mean Jerusalem mid-winter (DJF) temperatures are displayed in Figure 6. Furthermore, precipitation fluctuations during 219 220 the instrumental period are super-imposed on the temperature fluctuations. Although a 221 precipitation reconstruction is not the subject of this paper, the variations seen in Fig. 6 indicate the strong negative correlation between temperature and precipitation in the 222 EM in agreement with Striem, (1979), who found that a decrease in  $1^{\circ}$ C is equivalent 223 224 to an increase of 100mm in seasonal precipitation in Jerusalem. To verify this, we compared our reconstructed temperature series with independent historical documents 225 of Klein (1985). The results are in agreement with Striem, (1979); i.e., the winters of 226 1763/4 1771/2, 1772/3, 1784/5, 1827/8, 1828/9, 1844/5, 1845/6 which were winters 227 228 with higher than average reconstructed temperature were historically documented as 229 dry winters (Klein, 1985), while the winters 1796/7, 1797/8, 1806/7, 1835/6, 1849/50, 1861/2 with lower than average reconstructed temperature were documented as wet 230 231 (Klein, 1985). These findings also strengthen the accuracy of our reconstructed series. 232 It is worth referring to the period 1882-1904 known as the 'First Aliyah' (FA) or the 'Agricultural Aliyah' period. This term is used to describe a large immigration of Jewish 233 234 Zionists to Israel, ~25,000-35,000, who managed, for the first time, to establish an

agricultural community. It is suggested here that this period had favouring climate conditions for dry farming agriculture in the relatively hot and dry conditions of Israel, 236 237 i.e., below normal temperatures and above normal precipitation. Before the FA, the 238 rainy seasons of 1873/4 and 1877/8 had exceptional high precipitation amounts 239 (~1000mm/year in Jerusalem) as recorded in observations, on the same order of magnitude as 1991/2, which was the most exceptional rainy year on record, related to 240 241 Mt. Pinatubo eruption (Bookman et al., 2014). Furthermore, 1879/80, 1881/2, 1882/3 and 1883/4 were historically documented as extremely wet years (Klein, 1985). The 242 243 temperature reconstruction presented here suggests that a partially relief in the hardness, of high temperatures and lack of precipitation, for agriculture probably helped to the 244 success of this agricultural immigration. These conditions have not been seen since 245 246 1750. The warming trend of recent years is unprecedented in the last 264 years and is in line with other studies. 247

#### 249 5 Summary

The study reconstructed the mid-winter temperatures in Jerusalem since 1750, using data from multiple climate proxies. The hypothesis that Jerusalem is a good 251 252 representative of the EM climatic fluctuations is proved, based on comparison of the 253 representativeness scores of the nine longest records of instrumental surface 254 temperature in the EM. Also, the high spatial correlations between the temperature in Jerusalem at the 20th century and the surface temperature fields in the EM and in 255 Europe, shown through correlation maps, strengthen this conclusion. The suitability of 256 European station records to serve as highly resolved predictors for the average 257 258 temperature in Jerusalem is well demonstrated, suggesting they can be used to reconstruct the larger EM climate. Historical documents used pointed at periods of 259 extreme temperatures and therefore, validate the reconstructed values. 260

The warming trend of recent decades is unprecedented since 1750, in agreement with global and regional studies (IPCC 2013). It is suggested here that the period of the First Aliyah (FA) to Israel coincided with favouring climate conditions for dry faming agriculture in the relatively hot and dry conditions of Israel. These conditions have not been seen since 1750, and probably helped in its success.

# 269 Acknowledgements

| 270 | This work was supported in part by Grant from the European Science Foundation in          |
|-----|-------------------------------------------------------------------------------------------|
| 271 | the frame work of Mediterranean Climate Variability and Predictability, the Porter        |
| 272 | school of Environmental studies, Tel-Aviv University, Tel-Aviv, Israel and GLOWA -        |
| 273 | Jordan River Project funded by the German Ministry of Science and Education               |
| 274 | (BMBF), in collaboration with the Israeli Ministry of Science and Technology (MOST).      |
| 275 | Finally, we acknowledge the kind hospitality and helpful discussions of Juerg             |
| 276 | Luterbacher, Heinz Wanner, Elena Xoplaki and the entire klimet GIUB group at Bern         |
| 277 | University, Bern, Switzerland. Special thanks go to Mrs. Judith Lempert, the linguistic   |
| 278 | editor of the text.                                                                       |
| 279 |                                                                                           |
| 280 | References                                                                                |
| 281 | Alpert, P. Jin, F. and Kitoh, A.: The Projected Death of the Fertile Crescent. In:        |
| 282 | Norwine J, (ed) A World after Climate Change and Culture-Shift, Springer, 193-            |
| 283 | 204, 2013.                                                                                |
| 284 | Alpert, P. and Coauthors: Relations between climate variability in the Mediterranean      |
| 285 | region and the Tropics: ENSO, South Asia and African Monsoons, hurricanes and             |
| 286 | Saharan dust. in: Lionello P, Malanotte-Rizzoli P, Boscollo R (eds) The                   |
| 287 | Mediterranean Climate: an Overview of the Main Characteristics and Issues.                |
| 288 | Elsevier, 149-177, 2006.                                                                  |
| 289 | Alpert, P. and Neumann, J.: An ancient "correlation" between streamflow and distant       |
| 290 | rainfall in the near East. Journal of Near Eastern studies 48: 313-314, 1989.             |
| 291 | Bar-Matthews, M. Ayalon, A. Kaufman, A.: Late quaternary paleoclimate in the eastern      |
| 292 | Mediterranean region from stable isotope analysis of speleothems at Soreq cave,           |
| 293 | Israel. Quaternary Research 47: 155-168, 1997.                                            |
| 294 | Bitan, A. and Rubin, S.: Climatic atlas of Israel for physical and environmental planning |
| 295 | and design. Ramot Publishing, Tel Aviv, Israel, 1994.                                     |
| 296 | Bookman, R. Filin, S. Avni, Y. Rosenfeld, D. and Marco, S.: Possible connection           |
| 297 | between large volcanic eruptions and level rise episodes in the Dead Sea Basin.           |
| 298 | Quaternary Science Reiews 89: 123-128, 2014.                                              |
| 299 | Brazdil, R. Pfister, C. Wanner, H. Storch, H. V. and Luterbacher, J.: Historical          |
| 300 | climatology in Europe - The state of the art. Climatic Change 70: 363-430, 2005.          |

- Casty, C. Wanner, H. Luterbacher, J. Esper, J. and Bohm, R.: Temperature and
- precipitation variability in the European Alps since 1500. International Journal of
- climatology 25: 1855-1880, 2005a.
- Cook, E. R.: A time series analysis approach to tree ring standardization. Dissertation,
  The University of Arizona, 171 pp, 1985.
- Cook, B. I. Anchukaitis, K. J. Touchan, R. Meko, D. M. and Cook, E. R.
- Spatiotemporal drought variability in the Mediterranean over the last 900 years.
- Journal of Geophysical Research. doi: 10.1002/2015JD023929, 2016.
- Ellenblum, R.: Climate change and the decline of the Eastern-Mediterranean, 950-
- 1072AD. Cambridge university press, New York, 2012.
- Enzel, Y. Bookman, R. Sharon, D. Gvirtzman, H. Dayan, U. Ziv, B. and Stein, M.:
- Late Holocene climates of the near East deduced from Dead Sea level variations in
- modern regional winter rainfall. Quaternary Research 60: 263-273, 2003.
- Finne, M. Holmgren, K. Sundqvist, H. S. Weiberg, E. and Lindblom, M.: Climate in
- the eastern Mediterranean, and adjacent regions, during the past 6000 years—a
- review. Journal of Archaeological Science 38: 3153–3173, 2011.
- Frumkin, A. Magaritz, M. Carmi, I. and Zak, I.: The Holocene climate record of the salt
  caves of mount Sedom. Holocene 1:191-200, 1991.
- Hasanean, H. M.: Fluctuations of surface air temperature in the Eastern Mediterranean.
- Theoretical and Applied Climatology 68: 75-87, 2001.
- Hsu, C. W. Chang, C. C. and Ling, C. J.: A Practical Guide to Support Vector
   Classification, 2003. <u>http://www.csie.ntu.edu.tw/~cjlin/papers/guide/guide.pdf.</u>
- Accessed 12 January 2014
- IPCC Climate Change 2013: The Physical Science Basis. Contribution of Working
   Group I to the Fifth Assessment Report of the Intergovernmental Panel on Climate
- Change [Stocker, T. F. Qin, D. Plattner, G. K. Tignor, M. Allen, S. K. Boschung, J.
- Nauels, A. Xia, Y. Bex, V. and Midgley, P. M. (eds.)]. Cambridge University Press,
- Cambridge, United Kingdom and New York, NY, USA, 1535 pp. doi:
   10.1017/CBO9781107415324, 2013.
- Kelley, C. P. Mohtadi, S. Cane, M. A. Seager, R. and Kushnir, Y.: Climate change in
  the Fertile Crescent and implications of the recent Syrian drought. PNAS 112(11):
- **332 3241-3246**, 2015.
- Kistler, and co-authors. : The NCEP-NCAR 50 year reanalysis: monthly means CD
- **ROM** and documentation. BAMS 82: 247-268, 2001.