# Peer review of "Mid-winter (DJF) temperature reconstruction in"

_Climate of the Past, 2016_

## Referee Comment (RC1) · Anonymous Referee #1 · 27 Oct 2016

The manuscript by Hochman et al. presents a statistical reconstruction of winter temperature in Jerusalem at annual resolution since 1750, where the observational record extends back to the year 1865. The major problem I have with this study is that the reconstruction is based almost solely on long instrumental records of precipitation (and SLP) from stations at teleconnected sites in central and western Europe. Such teleconnections are prone to non-stationarity in teleconnection strength and sign, especially under boundary conditions different than today (e.g., during the Little Ice Age). Furthermore, the authors ignore a previous temperature reconstruction for the Near East/Middle East region that used a similar approach, but based on multiproxy network reconstructions from teleconnected sites outside the region (Mann, Clim. Change

2002), and basically shows a similar temperature evolution back to 1750, which limits the novelty of the study by Hochman et al. What the community actually needs are annually-resolved temperature reconstructions from proxy archives or documentary data from the Middle East region itself, which are not potentially affected by the non-stationarity of teleconnection strength and sign compared to those reconstructions that are based on remote sites that are located outside the region. Hochman et al. fail to demonstrate that the published tree-ring chronology from Jordan they use, which is their only predictor from the Middle East region itself, significantly improves their Jerusalem temperature reconstruction. Moreover, the same tree-ring chronology has been originally (Touchan et al., 1999) and recently used to reconstruct Middle Eastern drought/precipitation evolution back in time (Cook et al., Sci. Adv. 2015, Cook et al., J. Geophys. Res. 2016), which is not mentioned or discussed in the present manuscript. The observational T and PP records from Jerusalem are not new, and have been widely discussed in the Israeli and wider literature, including the cold-wet vs. warm-dry pattern, that might be time scale dependent (e.g., interannual opposite to decadal and multidecadal variability). The last two sentences of the abstract are not supported by the data/analysis, as there have been agricultural communities in this region of the Middle East prior to 1882-1904 and, in particular, the Jerusalem precipitation record does not extend beyond the year 1865. Given the overall style of this work (sometimes it reads like a master thesis, numerous dissertations are cited, although there might be more appropriate references in the international peer-reviewed literature), I am not convinced that this manuscript by Hochman et al. meets the criteria for publication in the journal Climate of the Past.

Detailed comments: Line 42-44: The information regarding what occurred 12,000 years ago is a little bit out of context here.

Line 50-54: The study of Mann (Clim. Change 2002) is missing here, and there have been some more high-resolution studies from a variety of archives (lakes, tree-rings, stalagmites, corals) in the Eastern Mediterranean region.

Line 81-83: This sentence highlights that the scope of this study is rather regional than large-scale.

Line 105-109: Give WMO station numbers, as there are 2 stations in Jerusalem, at least for rainfall. Which one was used in the present study?

Line 117: More specific information on the database where the proxy data has been downloaded is needed here (not just the standard ncdc/noaa web address). Furthermore, you need to cite the original references for the tree-ring chronologies used.

Line 120: Provide more information on the kind of historical information that has been used, and do not cite dissertations that are not accessible to the readers of Climate of the Past.

Line124-125: Which parameter did you sue (NCEP/NCAR reanalysis), 300 hPa GPH?, and why?

Line 125-126: It is odd to use the outdated Kaplan dataset for sea surface temperature in your analysis. There more recent SST products (HadISST, ERSST4) that are more appropriate to use.

Line 167-174: Give at least the periods of the individual records here (see also comment regarding Table 2).

178-179: What are your criteria for "high significant correlations"?

181-182: The NAO has at least some influence on eastern Mediterranean climate on interannual to decadal time scales (PP, T), with a rather complex spatial anomaly pattern in the eastern Med for PP. This should be at least mentioned or discussed here.

188-191: This is not new, and has been described elsewhere, and reflects at least partly the influence of the NAO on the Mediterranean and adjacent continental regions.

Line 196-200: What about the positive correlation over the eastern Med/Caspian Sea region.

[Figure]

Line 198-203: This is all well known, see publications by Eshel, Rimbu, Felis for the eastern Med/Middle East, and by Xoplaki for the larger eastern Med region.

Line 220-223: Is the correlation significant on both interannual and decadal time scales?

Line 226-231: Referring to a dissertation is not appropriate. You just present 14 cases out of more than 100 years for higher/lower than average winter T. What is your threshold here?

Line 237-241: Why is this notable here?

Line 241-242: Is this evident in the observational record as well (proof of concept)?

Line 245-246: You cannot say this, as you do not have a PP reconstruction back to 1750.

Line 250-265: These statements are not robustly supported by the data, and what about significance?

Fig. 2: These patterns bear similarities to the NAO/NAM pattern. Provide information on the significance of your spatial correlations. Why did you use subperiods for your analysis?

Table 1: The standard NCDC/NOAA website is not a sufficient reference here. Provide more information and cite original references for the proxy records used here.

Table 2: Show the individual records in a figure. Give their periods in Table 2. These records are likely relatively short or have gaps. How does this influence the overall scoring/representation procedure?

---

## Referee Comment (RC2) · Anonymous Referee #2 · 22 Nov 2016

This paper presents a statistical reconstruction of annually-resolved winter (DJF) temperature in Jerusalem since 1750. The statistical methodology is based on principal component regression, using both instrumental data (precipitation and sea level pressure) from stations in central and western Europe and high temporal resolution records of proxy data (tree ring chronologies from Jordan). New climatic records are welcome and indeed needed for the region. However, as Referee #1 pointed out, I have serious concerns about the novelty, methodology and lack of supporting proxies.

Major Comments:

1) The authors stated that reconstructed winter temperature in Jerusalem is a first comprehensive attempt. However, Mann (2002) reconstructed annual and seasonal

patterns of temperature back through the mid-18th century through a method that combines all of the information available in instrumental, historical, and proxy climate indicators. Therefore, I couldn't truly get the point that makes the reconstructed winter temperature in Jerusalem different from Mann (2002). If the authors think that their study differs from Mann (2002) in terms of methodology, proxy sources, etc., they must include a discussion/comparison on that.

Mann, M. E., 2002. Large-scale climate variability and connections with the Middle East in past centuries. Climatic Change, 55(3), 287-314.

2) Without doubt, the authors are aware of that the most prominent influence of large-scale forcing is NAO, however, other teleconnection patterns have been demonstrated to play a key role in characterizing the eastern Mediterranean hydroclimate variability such as NCP and EAWR. In the manuscript, the authors chose long-term precipitation and sea level pressure observations located in central and western Europe indicating a highly correlated relationship with the climate of region of interest, which can largely be linked to the NAO. However, recent studies highlighted that past (up to LIA and MCA) hydroclimate variability of the eastern Mediterranean region has been controlled not by NAO forcing alone and more importantly, the character of the NAO and its teleconnections have been non-stationary, indicating contrasting spatio-temporal trends and patterns in the Mediterranean region (e.g., Roberts et al., 2012). My concerns about the methodology arise here because the authors have used a limited geographical coverage of dataset in which the dominance of non-stationary components at high frequencies of the climate signal may take place. In parallel to Referee #1, I strongly believe that including and interpreting available high-resolution paleo-records would add a contribution to the body of knowledge aimed at understanding the uncertainties coming from non-stationary climate signals.

Roberts, N. et al., 2012. Paleolimnological evidence for an east-west climate see-saw in the Mediterranean since AD 900. Global and Planetary Change, 84-845, 23-34.

Other Comments:

1) Northern Hemisphere meteorological season of winter starts from December 1 to February 28. Therefore, I am confused with the expression of "Mid-winter (DJF)". I recommend the authors just use "winter" or "mid-wet season" or "mid-cold season".

2) Pg. 3, line 42-49: Considering the importance of the extreme hydrometerological events such as 2007-2010 drought in Syria mentioned here, it would be good to analyze and discuss the extreme climate characteristics of Jerusalem. For instance, the authors discussed Jerusalem as a representative of Eastern Mediterranean climate. Is this evaluation also valid in the extreme hydrometeorological events? I would recommend the authors here include some extreme climate indices to perform the representative analysis.

3) Pg. 3, line 50-58: Given the fact that the region is located in a transition zone and under the influence of by sub-tropical and mid-latitude circulations as well as by tropical intrusions, the authors should include some other observations and/or high-resolution paleo-records in the southern parts of the region (see Major Comment #2). Please refer to:

Felis, T. and Rimbu, N., 2010. Mediterranean climate variability documented in oxygen isotope records from northern Red Sea corals – A review. Global and Planetary Change, 71, 232-241.

4) Pg. 7, line 167-174: I have some doubts and questions here:

i) Do the stations have exactly the same temporal coverage? If not, the method applied between short time series and longer ones would be misleading in the choice of reference stations.

ii) Has the methodology applied to each individual monthly series of a tested station? Or at seasonal time scale?

iii) I couldn't get the exact representative scores (mean correlations/mean significance)

in Table 2. Some of the values are different from mean correlations/mean significance result. Please check it out or am I missing something?

iv) How about the representativeness of the extreme hydrometeorological events and interannual variability? (Please see Other Comments #2).

5) Pg. 7, line 197: Why 300mb geopotential height? Please briefly explain the physical/dynamical reason for that.

Figures and Tables:

The figures are generally of good quality, however the labels and legends are difficult to read in Figure 2. It would be good to add a topographic background in Figure 1. Please check Table 2 (Please see Other Comments #4).

―――――――――――――――

---

## Author Comment (AC1) · 21 Dec 2016

The manuscript by Hochman et al. presents a statistical reconstruction of winter temperature in Jerusalem at annual resolution since 1750, where the observational record extends back to the year 1865. The major problem I have with this study is that the reconstruction is based almost solely on long instrumental records of precipitation (and SLP) from stations at tele connected sites in central and Western Europe.

In response: Other proxies were checked (e.g., speleothems from Soreq cave, tree rings from other sites in Jordan and Cyprus). The final data base for the temperature reconstruction is after a feature selection procedure which took only the significantly correlated proxies to the winter temperature reconstruction. In our opinion this is the

best that can be done at this stage until other natural proxies will emerge in the future. In the revised version we will bring forth all the proxies from the studied region that were checked and found not significant for the reconstruction of Jerusalem's winter temperature. Such teleconnections are prone to non-stationarity in teleconnection strength and sign, especially under boundary conditions different than today (e.g., during the Little Ice Age).

In response: In figure 2 we show that teleconnections are stable at least for the 20th century looking at different periods. The periods we have looked upon are 1900-1990 and 30 year sub-periods (1900-1930, 1930-1960, 1960-1990).

Furthermore, the authors ignore a previous temperature reconstruction for the Near East/Middle East region that used a similar approach, but based on multiproxy network reconstructions from tele connected sites outside the region (Mann,Clim. Change 2002), and basically shows a similar temperature evolution back to 1750, which limits the novelty of the study by Hochman et al.

In response: In the revised version we will refer to the study of Mann (2002) and will add a discussion of the novelty of this study with respect to Mann (2002). The limitations of this previous work that justify the current study are as follows: 1) Mann uses instrumental temperature records to reconstruct the Mnet index. We show in figure 2 that the low relations between the temperature in Europe and in Jerusalem, whereas, precipitation and SLP have much stronger and stable relations. Furthermore, Mann's instrumental records stretch back only to 1854 while the records we have used extend back to 1750. We have chosen only instrumental records that extend back to 1750 and are statistically related to the winter temperature record in Jerusalem as shown in figures 2, 3, 4) Mnet is a gridded index which combines temperatures from 12 grid points over the EM. These grid points are located in different climatic regions, i.e., the subtropical desert and the temperate Mediterranean climates, as well as in a complex terrain area that includes the Mediterranean Sea, the coastal plain and the mountainous region. Our reconstruction focuses on a localized region of the EM (Jerusalem). This makes our reconstruction more robust for Jerusalem than an index averaged over a complex region. 3) Mann's seasonal reconstructions are for October to March (the "cold" season). This includes periods (especially in October November and March) that the study region is influenced by tropical intrusions and might have lower relations with Europe. Thus, our study focuses on the mid-winter (DJF) were strong relations with Europe do exist, as well as with the NAO. 5) As stated by the reviewer, proxy based records are sparse in the EM. Mann does not use any while we find that the Jordan tree chronology is related to the Jerusalem temperature record which was never tested before. In all simulations for choosing the most appropriate proxies the Dana reserve chronology, which is based on 17 tree chronologies, was found significant to the reconstruction of Jerusalem winter temperature. This will be clarified in the revised version. Furthermore, the contribution of other proxies was tested as well; such as speleothems from Soreq cave, and tree rings from Cyprus. The final data set presented includes only the significantly related proxies to the Jerusalem winter temperature 6) we use different verification statistics for our reconstruction in the form of the well-established cross validation method shown in figure 5 and table 4. These results show an improvement in the accuracy compared to Mann's results. 7) Our reconstruction includes the last period, from 2000 to present, which was not available during the publication of Mann (2002). 8) Finally, the reviewer suggests that a similar evolution of temperature is found in Mann (2002), therefore, our study reinforces the former study using different proxies. In our perspective, this is an important step forward that deserves publication.

What the community actually needs are annually-resolved temperature reconstructions from proxy archives or documentary data from the Middle East region itself, which are not potentially affected by the non-stationarity of teleconnection strength and sign compared to those reconstructions that are based on remote sites that are located outside the region. Hochman et al. fail to demonstrate that the published tree-ring chronology from Jordan they use, which is their only predictor from the Middle East region itself, significantly improves their Jerusalem temperature reconstruction. Moreover, the same tree-ring chronology has been originally (Touchan et al., 1999) and recently

used to reconstruct Middle Eastern drought/precipitation evolution back in time (Cook et al., Sci. Adv. 2015, Cook et al., J. Geophys. Res. 2016), which is not mentioned or discussed in the present manuscript.

In response: We have tried many combinations of proxies in order to reduce the target functions (e.g., RMSD, correlation etc.). The final proxies chosen give the best cross validation results we could reach. The Jordan tree ring chronology was always found to be an important proxy, contributing to the overall skill of the reconstruction model. We will demonstrate the importance of the Jordan tree ring chronology for the reconstruction of the Jerusalem temperature series in the revised version. Cook et al 2016 is mentioned in the text (line 77-80) and will be discussed more extensively in the revised version. Cook reconstructs summer (JJA) droughts with relation to winter and summer precipitation. Our study is focused on winter temperatures in a localized region. Furthermore, Cook et al. 2016 also use tele connected proxies in the form of tree rings as does Mann et al 2002. We use tele connected instrumental series along with local tree ring chronologies for our reconstruction. Their study reinforces the possibility of using tele connected sites for a local reconstruction as we further demonstrate in our paper. Cook et al. 2015 discusses projections into the future and not reconstructions therefore their study was not discussed in the paper.

The observational T and PP records from Jerusalem are not new, and have been widely discussed in the Israeli and wider literature, including the cold-wet vs. warm dry pattern, that might be time scale dependent (e.g., interannual opposite to decadal and multidecadal variability).

In response: Although the Jerusalem record was discussed before, a temperature reconstruction for this important site was not brought forth. Our study brings forth a winter temperature reconstruction using a combination of instrumental and natural proxies.

The last two sentences of the abstract are not supported by the data/analysis, as there

have been agricultural communities in this region of the Middle East prior to 1882-1904 and, in particular, the Jerusalem precipitation record does not extend beyond the year 1865. Given the overall style of this work

In response: Although there have been agricultural communities before 1882, there was no mass immigration to the region prior to 1882. We suggest that this might, at least partly, be related to climate variability and should be further checked. Although the precipitation record does not extend further back than 1865 the temperature reconstruction does. First we show that the relationship between temperature and precipitation is strong in the instrumental period (figure 6). This was already discussed by Striem (1977). Furthermore, using this relation, it is shown (figure 6) that before 1865 the winter temperatures were not as low as in the period 1882-1904 and since that period the combination of low temperatures and large amounts of precipitation for such a long period did not return.

(sometimes it reads like a master thesis, numerous dissertations are cited, although there might be more appropriate references in the international peer-reviewed literature),

In response: dissertations were used only where no peer reviewed articles were available. We will revise the manuscript with the latest updated references.

I am not convinced that this manuscript by Hochman et al. meets the criteria for publication in the journal Climate of the Past.

In response: Based on the arguments raised above, we believe that the revised version will merit publication in Climate of the Past Journal.

Detailed comments: Line 42-44: The information regarding what occurred 12,000 years ago is a little bit out of context here.

In response: The next sentence relates to the Fertile Crescent and the recent drought 2007-2010. The study region is located in the Fertile Crescent which is projected to

disappear at the end of the century. We believe that this strengthen the insight into the high climatic variability of this region. Line 50-54: The study of Mann (Clim. Change 2002) is missing here, and there have been some more high-resolution studies from a variety of archives (lakes, tree-rings, stalagmites, corals) in the Eastern Mediterranean region.

In response: The revised version will contain a rigorous discussion of Mann (2002) as described above, as well as other studies.

Line 81-83: This sentence highlights that the scope of this study is rather regional than large-scale.

In response: The study is a regional study highlighting teleconnections with Europe and the NAO which are large scale processes.

Line 105-109: Give WMO station numbers, as there are 2 stations in Jerusalem, at least for rainfall. Which one was used in the present study?

In response: The Jerusalem record is WMO 40184 and homogenized in the Israeli Meteorological Service. This data will be added in the revised version.

Line 117: More specific information on the database where the proxy data has been downloaded is needed here (not just the standard ncdc/noaa web address). Furthermore, you need to cite the original references for the tree-ring chronologies used.

In response: This will be added in the revised version.

Line 120: Provide more information on the kind of historical information that has been used, and do not cite dissertations that are not accessible to the readers of Climate of the Past.

In response: The historical record was not published in a peer reviewed journal, therefore the dissertation is the only source.

Line124-125: Which parameter did you use (), 300hPaGPH? And why?

In response: This is discussed in lines 196-203. "Figure 4 presents teleconnection patterns between Jerusalem mid-winter temperature and 300mb geopotential heights (taken from NCEP/NCAR realysis). A strong negative relationship is observed over central and western Europe. These findings along with the negative SLP relationship with Jerusalem mid-winter temperature (Figure 2) reinforce the statistical relationship between mid-winter precipitation in Europe and temperature in Jerusalem, with a dynamical one. European high pressure causes northerly winds in Israel which bring cold air from western Russia to the region, in a process governed by the phase configuration of Rossby waves (Ziv et al., 2006)." Following your question we will mention it earlier in the text, as needed.

Line 125-126: It is odd to use the outdated Kaplan dataset for sea surface temperature in your analysis. There more recent SST products (HadISST, ERSST4) that are more appropriate to use.

In response: It is possible to present a different data set if needed. This will be checked in the revised version, as well as the correlation with the Kaplan data set.

Line 167-174: Give at least the periods of the individual records here (see also comment regarding Table 2).

In response: The periods will be given in the revised version.

178-179: What are your criteria for "high significant correlations"?

In response: Figure 2 exhibits correlations of $>\pm0.6$ and the color regions are significant at the 95% level.

181-182: The NAO has at least some influence on eastern Mediterranean climate on interannual to decadal time scales (PP, T), with a rather complex spatial anomaly pattern in the eastern Med for PP.This should be at least mentioned or discussed here.

In response: The NAO was discussed extensively in the literature. We agree that it should be at least mentioned in the revised version and will do so.

188-191: This is not new, and has been described elsewhere, and reflects at least partly the influence of the NAO on the Mediterranean and adjacent continental regions.

In response: We agree that this reflects, at least partly, the NAO influence and references will be added. Nevertheless, we present it in order to establish the use of instrumental teleconnections as proxies for Jerusalem winter temperature, which was not done before.

Line 196-200: What about the positive correlation over the eastern Med/Caspian Sea region.

In response: Figure 4 shows the correlation over the Caspian Sea. The reason we do not discuss it here because the focus is on establishing the use of European data located in Western and central Europe. This correlation will be discussed in the revised version. It is probably related to the East Atlantic Western Russia (EA/WR) pattern, which is one of the three prominent patterns affecting Euro Asia throughout the year.

Line 198-203: This is all well known, see publications by Eshel, Rimbu, Felis for the eastern Med/Middle East, and by Xoplaki for the larger eastern Med region.

In response: We agree that this is well known, but brought forth here to establish the use of instrumental teleconnections for the reconstruction of Jerusalem winter temperature. Since it is well known it strengthens our hypothesis which the reviewer has criticized in the beginning.

Line 220-223: Is the correlation significant on both interannual and decadal time scales?

In response: This was not looked into and will be checked in the revised version.

Line 226-231: Referring to a dissertation is not appropriate. You just present 14 cases out of more than 100 years for higher/lower than average winter T. What is your threshold here?

In response: The years we present are the years which we have historical records to verify from Klein (1985). If we had more records we would have added them. We have looked at the peaks in the reconstructed temperature series with respect to the historical record.

Line 237-241: Why is this notable here?

In response: it is noted here since immigration does not start at one point of time. It takes time to understand that it is possible. We speculate that Jewish immigrants of the "First Aliyah" have heard of wet and cold years in the region like 1873/4 1877/8 and on this continuum, after a few wet cold years have decided to immigrate during the period 1882-1904. Verifying these historical facts is beyond the scope of this paper.

Line 241-242: Is this evident in the observational record as well (proof of concept)?

In response: We will add a section in the revised version. The precipitation series of Jerusalem does show that the specified years had above average precipitation amounts.

Line 245-246: You cannot say this, as you do not have a PP reconstruction back to 1750.

In response: This is stated based on the already established relation between temperature and precipitation (Striem 1979). It is shown that the period 1882-1904 was exceptional at least for the instrumental period. This study is the first to propose that the idea of mass immigration may, at least partly, be related to climatic changes.

Fig. 2: These patterns bear similarities to the NAO/NAM pattern. Provide information on the signiïficance of your spatial correlations. Why did you use subperiods for your analysis?

In response: The significance levels are stated in the legend of the figure. The color regions are significant at the 95% level. We have used sub periods in order to show that the relations are stable for western and central Europe, at least for the period that

we have data to verify these relations.

Table 1: The standard NCDC/NOAA website is not a sufficient reference here. Provide more information and cite original references for the proxy records used here.

In response: The data and details will be added in the revised version.

Table 2: Show the individual records in a figure. Give their periods in Table 2.

In response: This will be done in the revised version.

These records are likely relatively short or have gaps. How does this influence the overall scoring/representation procedure?

In response: The stations we checked are the longer instrumental stations in the Eastern Mediterranean. The correlations were checked on the 1930-1990 period which virtually no missing values are present. This will be discussed in the revised version.

---

## Author Comment (AC2) · 21 Dec 2016

This paper presents a statistical reconstruction of annually-resolved winter (DJF) temperature in Jerusalem since 1750. The statistical methodology is based on principal component regression, using both instrumental data (precipitation and sea level pressure) from stations in central and western Europe and high temporal resolution records of proxy data (tree ring chronologies from Jordan). New climatic records are welcome and indeed needed for the region. However, as Referee #1 pointed out, I have serious concerns about the novelty, methodology and lack of supporting proxies.

Major Comments: 1) The authors stated that reconstructed winter temperature in Jerusalem is a first comprehensive attempt. However, Mann (2002) reconstructed an-

nual and seasonal patterns of temperature back through the mid-18th century through a method that combines all of the information available in instrumental, historical, and proxy climate indicators. Therefore, I couldn't truly get the point that makes the reconstructed winter temperature in Jerusalem different from Mann (2002). If the authors think that their study differs from Mann (2002) in terms of methodology, proxy sources, etc., they must include a discussion/comparison on that. Mann, M. E., 2002. Large-scale climate variability and connections with the Middle East in past centuries. Climatic Change, 55(3), 287-314.

In response: Please see the response to reviewer 1 for the same comment.

2) Without doubt, the authors are aware of that the most prominent influence of largescale forcing is NAO, however, other teleconnection patterns have been demonstrated to play a key role in characterizing the eastern Mediterranean hydroclimate variability such as NCP and EAWR. In the manuscript, the authors chose long-term precipitation and sea level pressure observations located in central and western Europe indicating a highly correlated relationship with the climate of region of interest, which can largely be linked to the NAO. However, recent studies highlighted that past (up to LIA and MCA) hydroclimate variability of the eastern Mediterranean region has been controlled not by NAO forcing alone and more importantly, the character of the NAO and its teleconnections have been non-stationary, indicating contrasting spatio-temporal trends and patterns in the Mediterranean region (e.g., Roberts et al., 2012). My concerns about the methodology arise here because the authors have used a limited geographical coverage of dataset in which the dominance of non-stationary components at high frequencies of the climate signal may take place. In parallel to Referee #1, I strongly believe that including and interpreting available high-resolution paleo-records would add a contribution to the body of knowledge aimed at understanding the uncertainties coming from non-stationary climate signals. Roberts, N. et al., 2012. Paleolimnological evidence for an east-west climate see-saw in the Mediterranean since AD 900. Global and Planetary Change, 84-845, 23-34.

In response: Thank you for highlighting this important study. In our opinion no one climate reconstruction study can give the full picture. As climate scientists we can only present the data we have with its associated uncertainties, as presented in the text. The NAO is not the only climate index but it accounts for a large portion of the variability in temperature in the discussed region. Although Roberts et al. demonstrate interesting reconstructions, their reconstructions are based on lake sediments which do not capture inter-annual variability and are difficult to date. Furthermore, their data is not seasonally resolved. Summer and winter together in the same reconstruction obscure the signal. The comparison is done for lakes in Turkey and Spain both located in the mountains which are affected strongly by local factors. Nar Lake in Turkey is located $\sim$ 1000km to the North of Jerusalem in a mountainous region of Turkey.

Other Comments: 1) Northern Hemisphere meteorological season of winter starts from December 1 to February 28. Therefore, I am confused with the expression of "Mid-winter (DJF)". I recommend the authors just use "winter" or "mid-wet season" or "mid-cold season".

In response: We will consider changing the names in the revised version.

2) Pg. 3, line 42-49: Considering the importance of the extreme hydrometerological events such as 2007-2010 drought in Syria mentioned here, it would be good to analyze and discuss the extreme climate characteristics of Jerusalem. For instance, the authors discussed Jerusalem as a representative of Eastern Mediterranean climate. Is this evaluation also valid in the extreme hydrometeorological events? I would recommend the authors here include some extreme climate indices to perform the representative analysis.

In response: This would be an interesting idea to follow-up for this paper, but is beyond the scope of the mean temperature reconstruction.

3) Pg. 3, line 50-58: Given the fact that the region is located in a transition zone and under the influence of by sub-tropical and mid-latitude circulations as well as by tropical

intrusions, the authors should include some other observations and/or high-resolution paleo-records in the southern parts of the region (see Major Comment #2). Please refer to: Felis, T. and Rimbu, N., 2010. Mediterranean climate variability documented in oxygen isotope records from northern Red Sea corals – A review. Global and Planetary Change, 71, 232-241.

In response: Tropical intrusions in the form of the Red Sea Trough mostly affect the temperatures and precipitation of autumn and spring. This is why we focus solely on the mid-winter (DJF) in which air masses mostly propagate from the West. Felis and Rimbu strongly support our notion of using European precipitation records and reviews the connection with the NAO/AO oscillation for winter season for the last few centuries even in Red Sea corals. The resolution of this kind of proxy is not on the same order of tree rings and instrumental data.

4) Pg. 7, line 167-174: I have some doubts and questions here: i) Do the stations have exactly the same temporal coverage? If not, the method applied between short time series and longer ones would be misleading in the choice of reference stations.

In response: These are the long instrumental records for the region. The calculations were done on the 1930-1990 period were virtually no missing values are present. We will discuss this in the revised version.

ii) Has the methodology applied to each individual monthly series of a tested station? Or at seasonal time scale?

In response: seasonal time scale

iii) I couldn't get the exact representative scores (mean correlations/mean significance) in Table 2.Some of the values are different from mean correlations/mean significance result. Please check it out or am I missing something?

In response: The spatial representative score was developed so that High average correlations between pairs are high and the p value low the station gets a higher representative score. The average correlations and average significance levels are rounded numbers to two digits, therefore you don't get the exact representative score which is shown. We will improve the table so it would be clear in the revised version.

iv) How about the representativeness of the extreme hydrometeorological events and interannual variability? (Please see Other Comments #2).

In response: This could be interesting to check, but it was not done in this study.

5) Pg. 7, line 197: Why 300mb geopotential height? Please briefly explain the physical/dynamical reason for that.

In response: The reason we show this map is to explain the use of tele connected instrumental data for the reconstruction of Jerusalem. It is briefly explained in the text and can be further discussed in the revised version: "Figure 4 presents teleconnection patterns between Jerusalem mid-winter temperature and 300mb geopotential heights. A strong negative relationship is observed over central and western Europe. These findings along with the negative SLP relationship with Jerusalem mid-winter temperature (Figure 2) reinforce the statistical relationship between mid-winter precipitation in Europe and temperature in Jerusalem, with a dynamical one. European high pressure causes northerly winds in Israel which bring cold air from western Russia to the region, in a process governed by the phase configuration of Rossby waves (Ziv et al., 2006). "

Figures and Tables: The figures are generally of good quality, however the labels and legends are difficult to read in Figure 2. It would be good to add a topographic background in Figure 1. Please check Table 2 (Please see Other Comments #4).

In response: Thanks for these comments. We will improve the specified figures in the revised version.